

# A microbolometer-based far infrared radiometer to study thin ice clouds in the Arctic

Q. Libois[1], C. Proulx[2], L. Ivanescu[1], L. Coursol[1], L. Pelletier[1], Y. Bouzid[1],
F. Barbero[1], and J.-P. Blanchet[1]

[1]ESCER Centre, Department of Earth and Atmospheric Sciences, University of Quebec at
Montreal, Quebec, Canada
[2]Institut National d'Optique, 2740 Einstein Street, Quebec City, QC G1P 4S4, Canada

*Correspondence to:* Q. Libois (libois.quentin@uqam.ca)

**Abstract.** A far infrared radiometer (FIRR) dedicated to measure radiation emitted by clear and cloudy atmospheres was developed as a breadboard for the Thin Ice Clouds in Far InfraRed Experiment (TICFIRE) satellite project. The FIRR detector is an array of $80 \times 60$ uncooled microbolometers coated with gold black to enhance the absorptivity and responsivity. A filter wheel is used to select

atmospheric radiation in 9 spectral bands ranging from 8 to 50 $\mu$m. Calibrated radiances are obtained using two well-calibrated blackbodies. Images are acquired at a frame rate of 120 Hz, and temporally averaged to reduce electronic noise. A complete measurements sequence takes about 120 seconds. With a field-of view of $6°$, the FIRR is not intended to be an imager. Hence spatial average is computed over 193 illuminated pixels to increase the signal-to-noise ratio and consequently the detector

resolution. This results in an improvement by a factor of 5 compared to individual pixel measurements. Another threefold increase in resolution is obtained using 193 non-illuminated pixels to remove correlated electronic noise, leading an overall resolution of approximately 0.015 W m$^{-2}$ sr$^{-1}$. Laboratory measurements performed on well known targets suggest an absolute accuracy close to 0.02 W m$^{-2}$ sr$^{-1}$, which ensures to retrieve atmospheric radiance with an accuracy better than 1%.

Preliminary *in situ* experiments performed from the ground in winter and in summer on clear and cloudy atmospheres are compared to radiative transfer simulations. They point out the FIRR ability to detect clouds and changes in relative humidity of a few percent in various atmospheric conditions, paving the way for the development of new algorithms dedicated to ice cloud characterization and water vapor retrieval.

## 1    Introduction

During the Arctic polar night, the energy budget of the surface and atmosphere are mainly governed by longwave radiative fluxes (Overland et al., 1997). Of uttermost importance is the far-infrared (F-IR) spectral region ($\lambda > 15\mu$m), from which approximately 40% of the energy escaping the Earth originates (Harries et al., 2008), and where more than 60% of the atmospheric cooling of the dry and



cold Arctic atmosphere occurs (Clough et al., 1992). A refined comprehension of the Arctic climate
thus requires a thorough understanding of the radiative properties of the atmosphere in the F-IR. It
is well established that these radiative properties are highly sensitive to the amount of water vapor
(Turner and Mlawer, 2010; Bianchini et al., 2011) and to the physical properties of clouds (Curry,
1983; Maestri et al., 2014). However, these two major components of the hydrological cycle and their

radiative contributions remain poorly known in the Arctic winter. The radiative processes at stake
are indeed complex and still constitute active fields of research (Yang et al., 2005; Palchetti et al.,
2008; Delamere et al., 2010; Turner et al., 2012). In addition, accurate measurement of humidity and
cloud properties during the polar night is difficult (Cimini et al., 2009; Eastman and Warren, 2010;
Blanchard et al., 2014). For instance, decadal cloud cover trends are uncertain and the negative trend

observed by Wang (2003) using satellite data was not confirmed by surface observation reported
by Eastman and Warren (2010). The recent deployment of satellites dedicated to clouds remote
sensing, e.g. CALIPSO (Winker et al., 2003) and CloudSat (Stephens et al., 2002), has nevertheless
filled a gap in clouds observations at high latitudes (Delanoë and Hogan, 2010). It has, in particular,
highlighted the ubiquity of thin ice clouds (TICs) barely visible before (Grenier et al., 2009).

Although these TICs are characterized by small optical thickness ($\leq 5$), they have a critical radia-
tive impact. They indeed act as atmospheric radiators and their cooling efficiency is very sensitive
to their microphysical properties (Curry et al., 1996). If ice crystals become large enough to precip-
itate, this dehydrates the atmosphere which reduces the latter opacity and cools down the surface
and consequently the lower troposphere. This process, coined as 'Dehydration Greenhouse Feed-

back' by Blanchet and Girard (1994), has the potential to further cool the atmosphere and surface
(Blanchet and Girard, 1995). This provided a potential explanation for the observed negative trend
in Arctic surface temperatures observed in the late 20[th] century (Comiso, 2003; Wang, 2003). The
latter indeed remained unexplained by climate simulations which predicted a persistent warming
during winter due to the increase in anthropogenic greenhouse gases (Kahl et al., 1993). The radia-

tive impact of precipitating clouds raised the need to study in more details the TICs (Jouan et al.,
2012), and the F-IR proved to be a good candidate to observe those clouds (Blanchard, 2011) and
discriminate between precipitating and non-precipitating ones (Yang, 2003; Blanchard et al., 2009).
Such F-IR observations have already been used to study TIC at Eureka in the Arctic (Mariani et al.,
2012) and at Dome C in Antarctica (Palchetti et al., 2015).

With the intent to further investigate TICs formation and characteristics in the Arctic, a satellite
project sponsored by the Canadian Space Agency was initiated. The Thin Ice Clouds in Far InfraRed
Experiment (TICFIRE) mission (Blanchet et al., 2011) aims at filling a gap in remote sensing obser-
vation of the Earth in the F-IR (e.g. Maestri et al., 2014) with a special focus on TICs and water vapor
in the polar regions. The objective is to have a global picture of TICs and explore the impact of an-

thropogenic pollution on their physical properties (Grenier and Blanchet, 2010). This recent interest
for the F-IR, largely fostered by technology developments, gave rise to other satellite projects such





as REFIR (Palchetti et al., 1999, 2006) and CLARREO (Taylor et al., 2010). While many recent developments in the F-IR have made the choice of interferometers to get a hyperspectral picture of the Earth (e.g. Canas et al., 1997; Knuteson et al., 2004; Rochette et al., 2009; Bianchini and Palchetti, 2011), the TICFIRE detector is an uncooled microbolometer-based imaging radiometer. This choice is consistent with the constrained budget of the mission, with the emphasis put on obtaining 2D images of the clouds and radiance measurements.

On the way towards the TICFIRE mission, a breadboard has been developed to serve as a first experimental platform aimed at highlighting the potential of the future satellite mission. Here we present the far infrared radiometer (FIRR) prototype designed to measure radiation in 9 spectral bands ranging from 8 to 50 $\mu$m. The design and data acquisition procedure of the instrument are detailed in Sect. 2. Its radiometric performances are investigated in Sect. 3 and preliminary upward looking measurements taken on cloudy and clear sky scenes are presented in Sect. 4. These results are then discussed with regards to the TICFIRE mission requirements (Sect. 5).

## 2  Instrument design and data acquisition

This section presents the technical characteristics of the FIRR and the data acquisition procedure. The calibration algorithm used to convert raw measurements into calibrated radiance is also detailed.

### 2.1  Instrument design

The general principle of the FIRR is to measure radiation coming from the atmosphere, then to compare it to radiation measured on two calibration blackbodies (BBs) to deduce absolute atmospheric radiance (e.g. Shaw et al., 2005). Practically, the FIRR is made up of two separate devices that are electronically connected. The opto-mechanical device (OMD, Fig. 1a essentially comprises the scene selection mirror (hereinafter referred as SSM), the BBs and the optics enclosure (Fig. 1b which hosts the filter wheel assembly, the telescope and the detector. The instrument control device (ICD) contains most electronic components, the embedded computer controlling the complete system through a dedicated software and a hardware control panel. The FIRR design is compatible with both ground measurements and airborne operation.

#### 2.1.1  Opto-mechanical device (OMD)

The FIRR has two viewing ports, allowing up- and down-looking measurements with a field of view of 6°. Radiation coming from either direction or from the BBs is relayed to a filter wheel by a gold coated rotating mirror driven by a step motor (the SSM). The calibration enclosure containing the BBs and SSM is very similar in design to the AERI (Knuteson et al., 2004) and ASSIST (Rochette et al., 2009) instruments. The BB design has been thoroughly validated by the SSEC group at the University of Wisconsin (Best et al., 2003). The spectral reflectance of Z306 paint samples similar to





the BBs inner coating was measured by Surface Optics Corp. in the spectral range of the FIRR, from which the BB emissivity was deduced. The BBs temperature can be controlled from -30 to 60°C with 5 mK accuracy (Rochette et al., 2009). It can not be set at a temperature below the ambient temperature, though, because the BBs can only be heated.

The motorized filter wheel is used to select the appropriate spectral channel to be measured. It
currently hosts 9 1.25-inch diameter filters, as well as an opaque position and an open position. Six more positions are available on the wheel but currently unused. The transmittances of all filters were measured in the laboratory and are shown in Fig. 2. Their spectral characteristics are summarized in Table 1. All filters are interference filters except for the 30-50$\mu$m filter that is a mesh filter. The narrow field of view of the FIRR ensures near-normal incidence on the filters which is beneficial to
the angular uniformity of the transmittance.

The filtered radiation is relayed to the detector via a telescope based on a Schwarzschild configuration with two on-axis spherical mirrors. The use of reflective optical elements with an appropriate coating ensures high and uniform transmittance over all the spectral bands of the FIRR. The use of aluminium mirrors also ensures stability of the design with varying temperature. The system has
a f-number of 1.12. The detector positioned at the focal plane of the telescope is a 2D array (80 x 60 pixels) of 104-$\mu$m-pitch uncooled microbolometers developed at Institut National d'Optique (INO), similar to those used for the EarthCARE mission broadband radiometer (Wallace et al., 2009; Proulx et al., 2009). The pixels are coated with gold black to enhance absorbance and responsivity (Ngo Phong et al., 2015). The detector is embedded in a vacuum-sealable INO GEN5 package sealed
by a chemical vapor deposition diamond window to ensure broadband uniform transmittance of the signal to the detector. It is operated at 15°C, which offers the best compromise between detector sensitivity and thermal noise (both decreasing with temperature). Based on the telescope design, a circular-shaped array of ∼300 pixels (∼2.1-mm diameter) has an adequately unobstructed view to the scene with homogeneous illumination. Nevertheless, to avoid any edge effects, only 193 pix-
els are actually used in this study. The detector signal is recorded by the INO IRXCAM camera at 120 Hz.

The optics enclosure contains heaters and thermometers to ensure temperature control and stability during operation, which is a prerequisite to achieve optimal radiometric accuracy. Temperature and relative humidity in the calibration enclosure are monitored as well. The OMD temperature is
stabilized at 15°C to match detector temperature. When ambient temperature is above this value, the OMD temperature increases because cooling is not possible.

### 2.1.2   Instrument control device (ICD)

The ICD consists of an electronics rack that contains the BBs controller, the instrument temperature regulation system and the embedded computer. The latter hosts a dedicated software that allows



manual and automatic measurements. The system can be powered by a standard 110 V/60 Hz supply
power or from a 28 V derivative current for use on board of an aircraft.

## 2.2  Data acquisition and processing

### 2.2.1  Data format

A single FIRR measurement consists in a series of frames acquired at fixed SSM and filter wheel
positions. It is stored in a *.raw* file containing the 16-bits numeric counts of all single frames and
their corresponding times of acquisition as well as a header recording the SSM and filter wheel
positions. The FIRR can be operated in manual or automatic mode. In manual mode, the position
of the SSM and that of the filter wheel are user-supplied and the number of frames to be taken has
to be provided. In automatic mode, the sequence of measurements to be taken is written in an input
file. All measurements taken in one sequence are stored in a single directory. The temperature and
humidity records acquired during the sequence, as well as the camera and BBs parameters, are stored
in this same directory. An automatic sequence can be repeated in a loop as many times as desired.

### 2.2.2  Instrument operation

FIRR measurements can be acquired following two different modes: the 'fast' and the 'slow' se-
quences. For the 'fast' sequence, the SSM is fixed on a position and the filter wheel is rotated to
illuminate consecutively all filters. This is repeated for the two calibration BBs and then for the
scene. For the 'slow' sequence, the filter wheel position is maintained and the SSM is rotated to
view consecutively the BBs and the scene. This is repeated for all filters. Since the rotation of the
SSM takes approximately 7 s whereas rotating the filter wheel takes about 1 s, the 'fast' sequence,
which minimizes the number of mirror rotations, is approximately 2.5 times faster than the 'slow'
one. In nominal conditions (one scene view, 11 filter wheel positions and 100 frames), the 'fast'
sequence takes 2 min and the 'slow' 5 min.

Both sequences have advantages and drawbacks. The 'fast' sequence is best suited when scene
variations are expected at time scales of a few minutes. First because its higher sampling rate al-
lows a better detection of such variations. Second because all scene measurements are taken in an
approximate 45 s interval. Comparatively, the scene measurements are taken throughout the 5 min
of the 'slow' sequence, making it difficult to compare different spectral bands in case of scene vari-
ations. In terms of calibration quality, the 'slow' sequence is better because it minimizes the time
interval between scene measurement and calibration for a given filter. Conversely, there is an approx-
imate 45 s between scene measurement and calibration for the 'fast' sequence, which is sufficient
for the background signal to change significantly in case of variable OMD temperature. In this case,
the background temporal variation must be handled adequately (Sect. 2.2.3). Eventually, when the
temperature of the calibration enclosure significantly differs from that of the filter wheel, the filter



temperature can change when it lies on the optical path. This temperature change is minor for the
‘fast’ sequence because the filter remains only a few seconds facing the calibration enclosure. On
the contrary, this temperature change and the corresponding variations in the filter self-emission can
become significant for the ‘slow’ sequence because the filter remains much longer in measurement
position. This contribution can not be removed by the calibration procedure, making it particularly
critical. To summarize, the ‘fast’ sequence is more suited for ground-based measurements of cloudy
scenes or aircraft measurements. The ‘slow’ sequence is appropriate for stable atmospheric condi-
tions at temperature near FIRR internal temperature and laboratory experiments with well controlled
environment.

### 2.2.3 Calibration

The FIRR detector collects radiation from the scene, but also from all the emitting parts of the
instrument which are in its field of view (e.g. Montanaro et al., 2014). When cold targets like ice
clouds are measured, the signal coming from the scene usually corresponds to only a few percent of
the total signal. To compute the true radiance of the scene from the total signal, a 2-point calibration
is used. It assumes that the total signal $C$ is the sum of a background signal $B$ and a scene signal $S$
(Revercomb et al., 1988). The main assumption is that the scene signal is proportional to the scene
radiance $L^{\mathrm{scene}}$. It implies that for each spectral band, the optical transfer function of the FIRR (from
SSM to detector package window) and the response of the detector be linear with incident radiance.
The first condition is fulfilled by the system and the second is evaluated in details in Sect. 3.2. Under
these conditions, the raw signal $C_{i,\lambda}^{\mathrm{scene}}$ for each pixel $i$ and each spectral band $\lambda$ can be written:

$$C_{i,\lambda}^{\mathrm{scene}} = B_{i,\lambda} + G_{i,\lambda} L_{\lambda}^{\mathrm{scene}}, \tag{1}$$

where $C_{i,\lambda}^{\mathrm{scene}}$ is the average count over all frames. To determine the background signal $B_{i,\lambda}$ and
gain $G_{i,\lambda}$, measurements are taken on two sources with known spectral radiance, namely a hot and
an ambient BB (HBB and ABB). For the calibration BBs, Eq. 1 reads:

$$C_{i,\lambda}^{\mathrm{ABB}} = B_{i,\lambda} + G_{i,\lambda} L_{\lambda}^{\mathrm{ABB}} \tag{2a}$$

$$C_{i,\lambda}^{\mathrm{HBB}} = B_{i,\lambda} + G_{i,\lambda} L_{\lambda}^{\mathrm{HBB}}. \tag{2b}$$

The BBs radiances are computed after Planck function using BBs temperatures and spectral emis-
sivities, as well as the temperature within the BBs enclosure since it contributes to the measured ra-
diance. The radiance in each spectral band $L_{\lambda}^{\mathrm{BB}}$ is obtained from the convolution of the BB radiance
by the filter normalized spectral transmittance. The background signal and gain are then computed
as:





$$G_{i,\lambda} = \frac{C_{i,\lambda}^{\mathrm{ABB}} - C_{i,\lambda}^{\mathrm{HBB}}}{L_\lambda^{\mathrm{ABB}} - L_\lambda^{\mathrm{HBB}}} \tag{3a}$$

$$B_{i,\lambda} = \frac{C_{i,\lambda}^{\mathrm{HBB}} L_\lambda^{\mathrm{ABB}} - C_{i,\lambda}^{\mathrm{ABB}} L_\lambda^{\mathrm{HBB}}}{L_\lambda^{\mathrm{ABB}} - L_\lambda^{\mathrm{HBB}}}. \tag{3b}$$

Figure 3 shows typical 2D maps of gain and background signal retrieved from Eqs. 3. The background signal shows a strong spatial gradient inherent to the array fabrication process. The gain pattern does not show such a gradient. Since the radiance in a single band is generally within 5-25 W m$^{-2}$ sr$^{-1}$, the signal from the scene generally accounts for only $\sim 1\%$ of the total signal, highlighting the need to remove the background signal properly.

The previous calibration procedure relies on the questionable assumption that the background signal remains constant throughout a sequence. In fact, the background is characterized by low frequency variations that can be linked to 1/f noise of the detector pixels and temperature change of surfaces seen by the detector outside of the scene. These variations can be significant when the 'fast' sequence is used but are less so in the case of the 'slow' sequence as the time required to see the

BBs and scene for a given filter is shorter in the latter case. Thorough investigation of background variations also pointed out the existence of correlated noise sources in addition to random signal variations typical of Johnson noise. These higher frequency variations may be the signature of detector temperature change (thermoelectric cooler controller) and electronic noise, and appear to be similar for all pixels. Hence, a non-illuminated area of pixels is defined (Fig. 3), so that a non-

illuminated pixel is associated to each illuminated pixel. The difference between the illuminated and non-illuminated signals is computed for each illuminated pixel to remove part of the correlated noise and obtain corrected raw data:

$$C_{i,\lambda}^{\mathrm{scene,ill}} = B_{i,\lambda}^0 + B_\lambda^1 + G_{i,\lambda} L_\lambda^{\mathrm{scene}} \tag{4}$$

$$C_{j,\lambda}^{\mathrm{scene,non-ill}} = B_{j,\lambda}^0 + B_\lambda^1 \tag{5}$$

$$C_{i,\lambda}^{\mathrm{scene,corr}} = (B_{i,\lambda}^0 - B_{j,\lambda}^0) + G_{i,\lambda} L_\lambda^{\mathrm{scene}}, \tag{6}$$

where $B^0$ is the slowly varying contribution of the background, $B^1$ is the correlated noise, and $j$ is the non-illuminated pixel corresponding to the illuminated pixel $i$.

To correct for the low frequency variations of the background which are critical for the 'fast'

sequence, linear time dependence is assumed. Since the rate of change $r$ represents a new unknown, a third equation is used to complement Eqs. 2. The next available measurement performed on the ABB is thus used. The set of calibration equations becomes:





$$C_{i,\lambda}^{\text{ABB,corr}}(t) = \left[B_{i,\lambda}^0(t) - B_{j,\lambda}^0(t)\right] + G_{i,\lambda}L_\lambda^{\text{ABB}}(t) \tag{7a}$$

$$C_{i,\lambda}^{\text{HBB,corr}}(t+\Delta t_1) = \left[B_{i,\lambda}^0(t) - B_{j,\lambda}^0(t)\right] + r_{i,\lambda}\Delta t_1 + G_{i,\lambda}L_\lambda^{\text{HBB}}(t+\Delta t_1) \tag{7b}$$

$$C_{i,\lambda}^{\text{ABB,corr}}(t+\Delta t_3) = \left[B_{i,\lambda}^0(t) - B_{j,\lambda}^0(t)\right] + r_{i,\lambda}\Delta t_3 + G_{i,\lambda}L_\lambda^{\text{ABB}}(t+\Delta t_3), \tag{7c}$$

where $\Delta t_n$ is the time lapse with regards to the initial ABB measurement. The calibrated radiance is eventually computed as:

$$L_{i,\lambda}^{\text{scene}}(t+\Delta t_2) = \frac{C_{i,\lambda}^{\text{scene,corr}}(t+\Delta t_2) - \left[B_{i,\lambda}^0(t) - B_{j,\lambda}^0(t)\right] + r_{i,\lambda}\Delta t_2}{G_{i,\lambda}}, \tag{8}$$

where $\Delta t_2$ is the time lapse between the ABB and scene measurements.

Since the FIRR is not intended to be used as an imager, the calibrated radiances are averaged over all illuminated pixels, meaning that all pixels are treated equally independently of their responsivity. The averaging procedure can alternatively be performed on the corrected signal before applying the calibration procedure, which is similar to considering the illuminated area as a single large pixel. Since $G$ and $r$ are nearly uniform, Eq. 8 is almost linear and both methods lead to similar results. Here, we use the second averaging method because it is more time efficient.

### 2.2.4 Overall algorithm

The general workflow used to process the FIRR data in the present study is as follows:

1. For each individual measurement:

    a  compute the temporal average and standard deviation over all frames

    b  select valid pixels with standard deviation less than 2 counts

    c  compute the difference between illuminated and non-illuminated pixels for valid pairs

    d  compute the 2D average of the corrected signal

2. Repeat step 1 for all single measurements of a sequence

3. Perform calibration to obtain radiances

It is often convenient to convert radiances into brightness temperatures for the sake of compatibility with other studies and instruments. Here the brightness temperature $T_B$ is defined as the temperature that a perfect BB should have to emit the measured radiance $L_\lambda^{\text{scene}}$ in a given band. It is computed as:

$$\int_\lambda \mathcal{T}_\lambda(\lambda')\frac{2hc^2}{\lambda'^5}\frac{1}{e^{\frac{hc}{\lambda' k_B T_B}}-1}\mathrm{d}\lambda' = L_\lambda^{\text{scene}}, \tag{9}$$





where $\mathcal{T}_\lambda$ is the normalized transmittance in band $\lambda$, $\lambda'$ the wavelength, $k_B = 1.38 \times 10^{-23}$ J K$^{-1}$ is the Boltzmann constant, $h = 6.63 \times 10^{-34}$ J s is the Planck constant and $c = 3.0 \times 10^8$ m s$^{-1}$ is the speed of light in vacuum.

## 3  Radiometric characterization

The radiometric performances of the FIRR, including noise equivalent radiance (NER), measurement repeatability, linearity of the detector and radiometric accuracy, are investigated based on laboratory experiments. For this, the ABB was left at ambient temperature ($\sim$25°C) and the HBB was set at 55°C. Two supplementary BBs facing the zenith and nadir ports were used (Fig. 4). These

BB cavities are identical to the FIRR calibration BBs. The zenith BB (ZBB) was controlled like the FIRR calibration BBs. The nadir BB (NBB) was immersed in a water-glycol bath, allowing to control its temperature from -27 to 55°C at 1 mK resolution. Nitrogen, at the bath temperature, is injected above the NBB to avoid condensation and contamination of the measurements by water vapor.

### 3.1  Instrument resolution

The detector performances are explored in terms of NER, which is a measurement of its radiometric resolution. NER is given by the ratio of the signal standard deviation over 120 frames (1 s) to the average gain:

$$\mathrm{NER}_\lambda = \frac{\sigma_\lambda}{G_\lambda},$$  (10)

where $G_\lambda$ is computed as:


$$G_\lambda = \left| \frac{C_\lambda^{\mathrm{HBB,corr}} - C_\lambda^{\mathrm{ABB,corr}}}{L_\lambda^{\mathrm{HBB}} - L_\lambda^{\mathrm{ABB}}} \right|.$$  (11)

Figure 5a shows the signal standard deviation over 120 frames as a function of the number of illuminated pixels over which the signal is averaged. The standard deviation decreases with the number of pixels used as a result of spatial averaging. It drops to 0.4 counts when 100 pixels are used and

reaches approximately 0.28 counts when all the illuminated pixels are used. The standard deviation is practically independent of the spectral band. It also appeared unchanged by the correction with non-illuminated pixels. The gain is shown in Fig. 5b for all spectral bands. It varies from one band to another from -16 to -28 counts W$^{-1}$ m$^2$ sr, because the gold black absorbance, hence the responsivity of the detector, is wavelength dependent (Ngo Phong et al., 2015). Also, spectral transmittance

varies from one filter to another. The 22.5 – 27.5 $\mu$m and 30 – 50 $\mu$m bands have the lowest transmittances (Fig. 2), which explains the low gain absolute values for these bands. As a result, the NER ranges from approximately 0.01 to 0.02 W m$^{-2}$ sr$^{-1}$ depending on spectral band and number of



pixels used. To illustrate the resolution of the FIRR in terms of brightness temperature, the NER is converted into noise equivalent temperature difference (NETD, e.g. Niklaus et al., 2008), that is

the temperature increase a perfect BB should experience for its radiance change to equal the NER. Figure 6 shows the variations of NETD with BB temperature for a NER of 0.01 W m$^{-2}$ sr$^{-1}$. The NETD decreases with temperature, the decrease being more pronounced for thermal infrared (T-IR, $5\ \mu$m $< \lambda < 15\ \mu$m) bands than for F-IR bands. At 20°C, the NETD ranges from 50 to 200 mK, whereas at -100°C it ranges from 200 to 650 mK. This is consistent with other radiometric systems

(e.g. Legrand et al., 2000). Note that at lowest temperatures, NETD is minimum for F-IR bands because the maximum of the Planck function shifts towards longer wavelengths.

The NER accounts for high-frequency noise at the time scale of one frame but noise also affects the signal at lower frequency. To estimate this lower frequency noise and its impact on the repeatability of the measurements, the difference between the signal acquired on ABB and HBB with the

$7.9 - 9.5\ \mu$m filter is computed for 40 consecutive sequences, which corresponds to more than 4 hours of measurements. This procedure is performed for corrected and non-corrected data and the results are shown in Fig. 7. The signal is fairly stable throughout the period of measurements which highlights the thermal stability of the system and BBs over this period. The standard deviation of uncorrected data is 0.57 counts but it drops to 0.17 counts when the correction is applied. This is

less than the standard deviation along 120 frames, which proves the very high repeatability of FIRR measurements and the benefit of using non-illuminated pixels.

**3.2   Instrument linearity**

The 2-point calibration procedure (Sect. 2.2.3) relies on the assumption that the contribution of the scene to the total signal delivered by the detector is proportional to the radiance of the source (Eq. 1).

To validate this critical assumption, the NBB was used as a well-defined source whose radiance is varied. For this, the temperature of the water-glycol bath was set successively to -27, -21, -15, -8, 0, 5, 15, 35 and 55°C, each temperature step lasting 75 min. FIRR measurements were taken continuously using the 'slow' sequence on ABB, HBB, nadir and zenith views with 200 frames to ensure optimal calibration. The corrected signal was computed for all filters using a restricted illuminated area of 69

pixels. This restriction is necessary because the NBB footprint on the detector is smaller than that of the calibration BB due to its larger distance to the SSM (Fig. 4). For the subsequent analysis, only the measurements taken for sufficiently stable NBB temperature are used. The threshold chosen is such that the standard deviation over one minute must be less than 2.5 mK. Practically, between 9 and 12 sequences met this criterion at each temperature step. The average NBB spectral radiances

are computed for each step. The difference between the average corrected signal on the NBB and



HBB is then computed. Since the temperature of the HBB is constant throughout the experiment, this difference is expected to increase linearly with NBB radiance:

$$C_\lambda^{\text{NBB,corr}} - C_\lambda^{\text{HBB,corr}} = G_\lambda \left( L_\lambda^{\text{NBB}} - L_\lambda^{\text{HBB}} \right). \tag{12}$$

Figure 8 shows this difference as a function of NBB radiance. In the temperature range explored, there is no evidence of non-linearity, which supports the calibration procedure used. The standard deviation over data acquired at the same temperature step ranges from 0.07 to 0.5 counts, which again gives an estimate of the long term stability of the system. The standard deviation of the difference between measurements and the regression line does not exceed 0.06 counts.

## 3.3   Instrument accuracy

The accuracy of the FIRR quantifies its capability to retrieve the absolute radiance of a target, whether the latter is cooler or warmer than both black bodies or when its temperature lies in between. To this end, the FIRR algorithm is used to retrieve the NBB radiance in the laboratory. Figure 9a shows the difference between retrieved and theoretical radiances for all sequences. The difference
does not exceed $0.08$ W m$^{-2}$ sr$^{-1}$, except for the band $30 - 50$ $\mu$m. The latter also exhibits a significant trend, with overestimation of the radiance at temperatures lower than ambient temperature and underestimation beyond. This trend is also observed for the other bands, but less pronounced. We hypothesize that this bias results from the difference in the geometrical positions of the NBB versus the calibration BBs (Mariani et al., 2012). In fact, the difference in optical path results in
lower air transmittance from the NBB to the SSM than from the calibration BBs to the SSM, which partly masks the actual temperature of the NBB. Radiative transfer calculations indeed show that the transmittance of 10 cm of air at $25\%$ relative humidity and $26°$C is only $88\%$ in the band $30 - 50$ $\mu$m while it is more than $97\%$ for all the others. To check this hypothesis, the retrieval algorithm is applied to the ZBB, whose distance to the SSM is much closer to that of the calibration BBs and
whose temperature varies within $25 - 55°$C. An illuminated area of 109 pixels is used and Fig. 9b shows differences less than $0.02$ W m$^{-2}$ sr$^{-1}$ for most bands, which is very close to the resolution of the FIRR. Since the uncertainty on the theoretical NBB radiance (less than $0.001$ W m$^{-2}$ sr$^{-1}$) is much less than the obtained difference, the value $0.02$ W m$^{-2}$ sr$^{-1}$ provides a good estimate of the FIRR accuracy.

# 4   Preliminary *in situ* experiments

Since the FIRR is aimed at remotely sensing clouds and atmospheric properties such as water vapor content, here three preliminary *in situ* experiments are presented. Two of them were performed in the core of winter in Québec City (Canada), one at night and the other in daylight conditions. The third experiment was performed during summer in Montréal (Canada). Those measurements are



compared to radiative transfer simulations performed with MODTRAN (Berk et al., 1987) in order
to assess the FIRR performances and explore its ability to characterize clouds.

## 4.1   Materials and methods

### 4.1.1   FIRR measurements

The FIRR measurements were acquired from the ground, using the zenith port of the instrument.
The 'fast' sequence was used and all data were acquired with 100 frames per measurement. The first
experiment was performed at INO facilities (46.80° N, 71.31° W) on 21 February 2015, from 00:27
to 00:50 UTC. The night was extremely clear and the synoptic situation characterized by a well set-
tled cold front bringing cold polar air (near surface temperature around -15°C), so that the conditions
were somehow similar to those encountered in early spring in the Arctic. The second experiment was
performed on 21 February 2015 from 13:30 to 20:00 UTC, again at INO facilities (Fig. 10), with
2 m air temperature around -10°C. The sky was uniformly overcast in the morning with the solar
disc still visible and very light precipitation. The cloud cover progressively vanished from 15:20 to
17:30 with almost clear sky at 16:00. From 18:00, the cloud cover became uniform again and was
visually thicker than in the morning. The solar disc was no longer visible. Light precipitation started
370    at 19:45. For these two experiments, the ABB and HBB temperatures were set at 15 and 50°C. The
last experiment was performed on 2 July 2015 on UQAM campus (45.51° N, 73.56° W) and lasted
24 hours. The air temperature varied between 15 and 25°C and relative humidity was around 30%.
The HBB was set at 55°C and the ABB was left uncontrolled at ambient temperature. The sky was
mostly clear in the first evening with sparse shallow cumulus appearing occasionally in the field of
view of the FIRR. During the night, consistent clouds (most likely stratocumulus) with larger hori-
zontal extension formed and remained from 7:00 to 10:00 UTC. The second day was similar to the
previous evening with sporadic occurrence of thin cirrus clouds.

### 4.1.2   MODTRAN simulations

The radiative transfer simulations were performed with MODTRAN v.5.3 (Berk et al., 1987). The
vertical profiles of pressure, temperature, humidity and ozone are user-defined. For the winter ex-
periments, these profiles were obtained from the closest 6-hourly ECMWF ERA-Interim reanalyses
(Dee et al., 2011). For the summer experiment, the hourly resolution RUC reanalyses from NOAA
(http://ruc.noaa.gov) were used. All other gas concentrations were set to default values for subarctic
winter (winter experiments) and midlatitude summer (summer experiment). Background aerosols
concentrations were set to rural value for winter experiments and urban values for the summer ex-
periment. Clouds are defined by a single layer homogeneous in ice and water content, whose optical
depth is prescribed. The spectral extinction and absorption coefficients of ice clouds are taken from
the parameterization of cirrus clouds of Yang et al. (2005). The model is run at 1 cm$^{-1}$ spectral res-



olution and multiple scattering is accounted for using the discrete ordinates method (Stamnes et al.,
1988) with 32 beams.

## 4.2 Results

### 4.2.1 Winter clear-sky experiment

The ERA-Interim vertical profiles of temperature and humidity at 00:00 are shown in Fig. 11. The
temperature close to the surface was -15°C and the air was relatively dry, except for the first 2 km. No
clouds were observed during the measurements during which eleven FIRR sequences were acquired.
Figure 12 shows the radiances measured with the FIRR along with the results of MODTRAN sim-
ulations. The sensitivity of the simulations to atmospheric inputs was estimated by adding a $\pm 1$ K
vertically uniform offset to the temperature profile or by perturbing by $\pm 10\%$ the specific humidity.
   The standard deviation of the measured radiances is less than 0.02 W m$^{-2}$ sr$^{-1}$ for all bands ex-
cept for the $30 - 50$ $\mu$m band, which is close to the resolution obtained in the laboratory (0.015 W m$^{-2}$ sr$^{-1}$).
This confirms very stable clear sky conditions during the experiment and provides an assessment of
FIRR measurements quality in environmental conditions. In particular, the presence of clouds, even
very thin, would have resulted in significant variations of the radiance in the $10 - 12$ $\mu$m band. The
fact that simulations and measurements coincide so well in all bands suggests that the atmospheric
reanalysis, the model and measurements are all consistent and of good quality. This provides a suc-
cessful radiative closure experiment for clear-sky conditions (e.g. Delamere et al., 2010; Fox et al.,
2015). The high sensitivity of the simulations to specific humidity in F-IR bands (except $30 - 50$ $\mu$m
that is saturated) shows that the humidity profile must be precise with more than 5% accuracy. It also
points out that the FIRR resolution is sufficient to discriminate between humidity profiles that differ
of a few percent in a very dry atmosphere. The simulations nevertheless show an apparent negative
bias in the F-IR, which is likely to be due to an erroneous water vapor profile. On the contrary, the
results of the T-IR bands between 10 and 14 $\mu$m, which are sensitive to the whole atmospheric pro-
file, ensure that the temperature profile is correct. The very low radiance measured in the $10 - 12$ $\mu$m
band, which corresponds to a brightness temperature of -125°C, matches very well the simulation.
It proves the quality of the calibration procedure, even when radiance is extrapolated far below the
calibration BBs radiances. As for the observed bias in the $7.9 - 9.5$ $\mu$m band, it can hardly be ex-
plained by errors on the water vapor or temperature profiles. Based on a series of tests, it is rather
likely the result of erroneous profiles of aerosols which were chosen somehow arbitrarily. Overall,
these results are very encouraging for further investigation of the water cycle in the Arctic.

### 4.2.2 Winter cloudy-sky experiment

Figure 13 shows the variations of the brightness temperature in all FIRR bands during the series of
measurements taken on 21 February 2015. Before 14:30 and after 18:00, all temperatures are within





a range of approximately 10°C, which is the signature of a cloud cover with sufficient optical depth. In between, the brightness temperatures show great variations as a result of cloud cover variations because the atmosphere becomes more transparent. These variations are visible in all bands except the $30 - 50 \, \mu$m band because the latter is saturated by the atmosphere emission below the clouds and does not actually see the clouds. Around 16:10, the sky is nearly perfectly clear and the brightness temperatures reach a minimum.

The radiances measured by the FIRR in clear sky and cloudy conditions were compared to MODTRAN simulations. For the simulations, the ERA-Interim profiles at 18:00 were used with an ice cloud layer lying between 2.3 and 3.8 km. A series of simulations was performed, with cloud optical depth $\tau$ ranging from 0.1 to 40 and cloud particles effective diameter $d_{\text{eff}}$ ranging from 3 to 250 $\mu$m. The 16:10 FIRR measurement (hereafter referred as clear sky measurement) is shown in Fig. 14. The corresponding MODTRAN simulation that best fits the measurements is shown as well. The latter corresponds to an optical depth of 0.3 and an effective diameter of 20 $\mu$m, which minimizes the mean square deviation with regards to the observations (Blanchard, 2011). The 18:20 FIRR measurement (hereafter referred as cloudy sky measurement) is also shown in Fig. 14 along with its corresponding optimal simulation, obtained for $\tau = 12.5$ and $d_{\text{eff}} = 40 \, \mu$m. The changes in atmospheric radiances observed with the FIRR are very well captured by the simulations in all bands. FIRR radiance increase in the $30 - 50 \, \mu$m band for the cloudy sky is not simulated because a single ERA-Interim profile is used for both simulations while near surface temperature actually increased between both measurements. The retrieved optical depths are consistent with cloud cover visual observations.

The optimization procedure applied to the clear and cloudy sky cases was applied to all measurements taken from 16:10 to 20:30. The time series of the retrieved optical depth and effective diameter is shown in Fig. 15. This highlights the increase of optical depth with a marked transition around 18:00 as expected from the brightness temperatures. The effective diameter is very noisy and can hardly be compared to visual observations. The variations nevertheless seem more consistent after 18:15 than before where the variations are physically irrelevant. The increase after 19:50 is also consistent with the observation of light precipitation at the end of the experiment. A CALIPSO (Winker et al., 2003) overpass at 17:53, 33 km East of the measurement site, provides useful data for evaluation of the FIRR. Lidar profiles show a cloud with $\tau = 3.32$ and the retrieval algorithm based on IR radiances suggests that $d_{\text{eff}} = 25 \, \mu$m. These values are shown in Fig. 15 and are consistent with the values retrieved by the FIRR. It seems that the optical depth retrieval works better for thin clouds while effective diameter retrieval is more consistent for thick clouds. These results are in agreement with the theoretical findings of Yang (2003). A similar method was used by Palchetti et al. (2015) to retrieve ice cloud properties on the Antarctic Plateau.



### 4.2.3    Summer experiment

The third experiment presented in this paper was performed in conditions very different from the previous ones with the intent to characterize the FIRR performances in relatively warm and humid

atmospheres. Figure 16 shows the time series of brightness temperatures for this 24 hr experiment. Sky pictures were taken every 30 s to provide qualitative information about the scene observed by the FIRR. The T-IR bands show large variations corresponding to clouds crossing the field of view of the FIRR. Narrow peaks at the beginning and at the end of the series correspond to small scale cumulus while the broader peak around 8:00 corresponds to the presence of a larger scale

stratocumulus. The narrow peak around noon is concomitant with the presence of a cirrus. F-IR bands show much smaller variations because the signal from the clouds is mostly attenuated by the warm and humid atmosphere below. Clear sky periods (e.g. 2:00 – 5:00) show the high repeatability of the measurements characterized by a resolution of $0.1 - 0.2$ K in all bands except the $30 - 50$ $\mu$m band. The latter exhibits the noisiest signal because it is sensitive to the BB enclosure temperature

and humidity variations rather than actual atmospheric radiance variations.

Figure 17 shows the measured spectral brightness temperatures for 4 different sky conditions with nearly constant temperature and water vapor profiles according to RUC reanalysis. The brightness temperature increases much more in the T-IR than in the F-IR pointing out the relative lack of interest of F-IR measurements in mid-latitude summer atmospheres. The MODTRAN simulation based on

RUC profiles at 6:00 perfectly matches the observations for clear-sky conditions, proving the quality of FIRR measurements in these singular conditions.

## 5    Discussion and conclusions

The radiometric performances of the FIRR were characterized through laboratory and *in situ* experiments and both approaches show very consistent results. With a value close to 0.02 W m$^{-2}$ sr$^{-1}$,

the FIRR radiometric resolution is much lower than the 0.1 W m$^{-2}$ sr$^{-1}$ required for the TICFIRE mission (Blanchet et al., 2011). The current resolution is nevertheless the result of extensive data processing which takes advantage of pixels spatial averaging and use of non-illuminated pixels. As the TICFIRE is meant to be an imager, spatial averaging will be limited. However, under the ergodicity hypothesis, temporal averaging can provide results similar to spatial averaging but reduces

acquisition rate and also the spatial resolution of the imager. All in all, a compromise will have to be made at the processing level between spatial and radiometric resolution, depending on the main objective of the TICFIRE mission. The current FIRR resolution is sufficient to detect very small humidity variations in a dry atmosphere and could provide water vapor columns retrieval with reduced uncertainties compared to other instruments like AIRS (Tobin et al., 2006) or REFIR-PAD (Bian-

chini et al., 2011) that provide 5-10% uncertainties on water vapor profiles. Comparison of FIRR



retrieved clouds properties with CALIPSO data also suggests that the FIRR is an adequate tool to detect and characterize ice clouds, which is its primary mission.

Nevertheless, the latter point could not be evaluated in more details in this study due to the lack of adequate data. The retrieval algorithm used in this study and based on a least square method is
meant to be a preliminary illustration of the FIRR capabilities. It does not pretend to reach a specific level of accuracy. A rigorous evaluation would require collocated profiles of clouds properties and radiosoundings as presented in e.g. Blanchard (2011). In addition, the impact of cloud particles size distribution, shape and aspect ratios was not investigated here while these parameters are known to be critical factors for clouds radiative properties (Chiriaco et al., 2004; Baran, 2007; Maestri et al.,
2014). A complete radiative closure experiment in the F-IR in cloudy conditions thus appears much more complicated than the experiment presented here for clear sky conditions. The development of an appropriate retrieval algorithm is the next step for FIRR applications with expected instrument deployment at Eureka, Nunavut (Canada) and Barrow, Alaska (USA) in dedicated experimental sites in the next years. At these very cold Arctic locations, the data quality is expected to be better than in
the present paper because the difference between ABB and sky temperatures will be less than in the present paper, enabling better calibration. Also, the data deterioration due to reduced transmittance of the air path between the calibration BBs and the FIRR detector should be much less in such cold and dry conditions with probable positive impact on the FIRR sensitivity in the $30 - 50$ $\mu$m band.

The FIRR novelty largely lies in its ability to probe radiation in the F-IR, an underexplored re-
gion of the Earth spectrum. The measurements presented in this paper already point out the FIRR capabilities in this remote spectral region. Indeed, with a sensitivity beyond 20 $\mu$m nearly as high as in the T-IR, the radiometric performances are very satisfying and promising for future detection and characterization of ice clouds in the Arctic. Further effort will be put on the development of an appropriate retrieval algorithm for cloud properties that will be evaluated with regards to other
reference instruments. These preliminary nevertheless results represent a substantial step toward the TICFIRE mission.

**Data availability**

The data used in this study are available upon request from the authors (libois.quentin@uqam.ca).

*Author contributions.* Q. Libois, L. Ivanescu, L. Coursol, L. Pelletier and F. Barbero participated to the lab-
oratory and *in situ* experiments. Q. Libois and C. Proulx developed the calibration algorithm. Q. Libois and Y. Bouzid performed the radiative transfer simulations. Q. Libois prepared the manuscript with contributions from the other authors.

*Acknowledgements.* This research was funded jointly by the Canadian Space Agency (CSA) through the FAST program and by NETCARE, the Canadian Network on Climate and Aerosols through the Climate Change and



Atmospheric Research (CCAR) program at NSERC. The authors are grateful to CSA for providing resources
and logistic support on the development of the FIRR instrument. We are very grateful to LR Tech Inc. for their
support during the FIRR laboratory characterization performed in their facilities.



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



**Table 1.** Filters spectral characteristics.

| Band number | Spectral range ($\mu$m) |
| --- | --- |
| 1 | 7.9 – 9.5 |
| 2 | 10 – 12 |
| 3 | 12 – 14 |
| 4 | 17 – 18.5 |
| 5 | 18.5 – 20.5 |
| 6 | 17.25 – 19.75 |
| 7 | 20.5 – 22.5 |
| 8 | 22.5 – 27.5 |
| 9 | 30 – 50 |

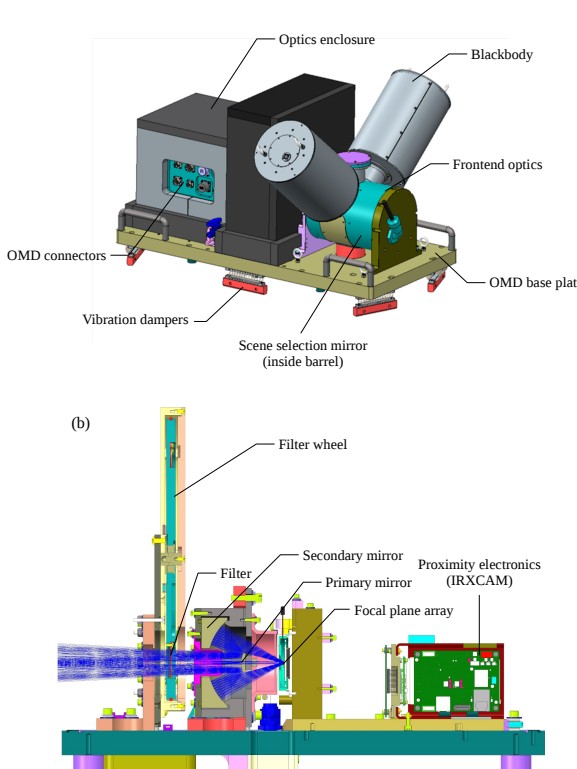

**Figure 1.** (a) Overview of the FIRR opto-mechanical device. The base plate dimensions are $90 \times 44$ cm. (b) Section view of the optics enclosure.



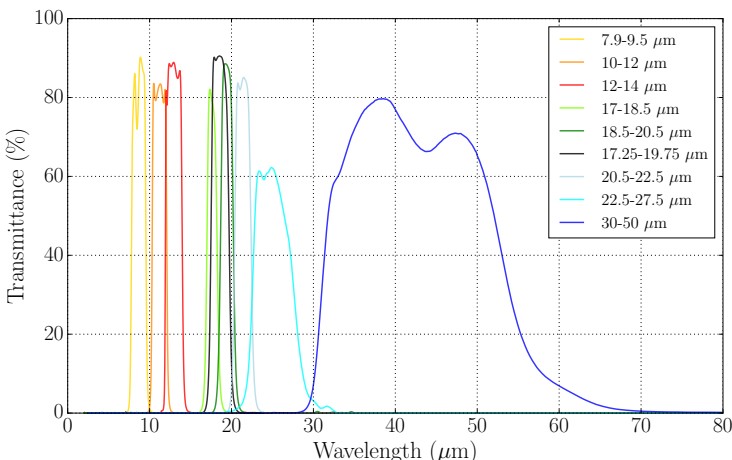

**Figure 2.** Spectral transmittance of the 9 filters of the FIRR.

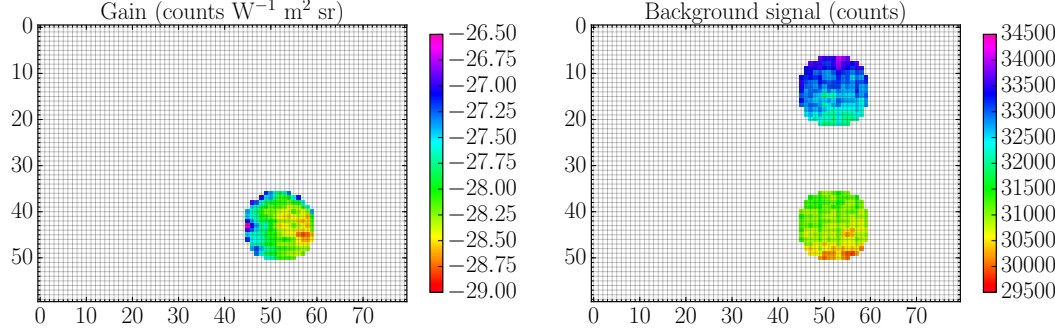

**Figure 3.** 2D maps of gain and background signal for filter 7.9-9.5$\mu$m for measurements acquired in the laboratory with $T^{HBB} = 55°C$ and $T^{ABB} = 24.5°C$. Only the 193 pixels used for data analysis are colored. On the right panel, the upper circle shows the non-illuminated area used for the calibration procedure.



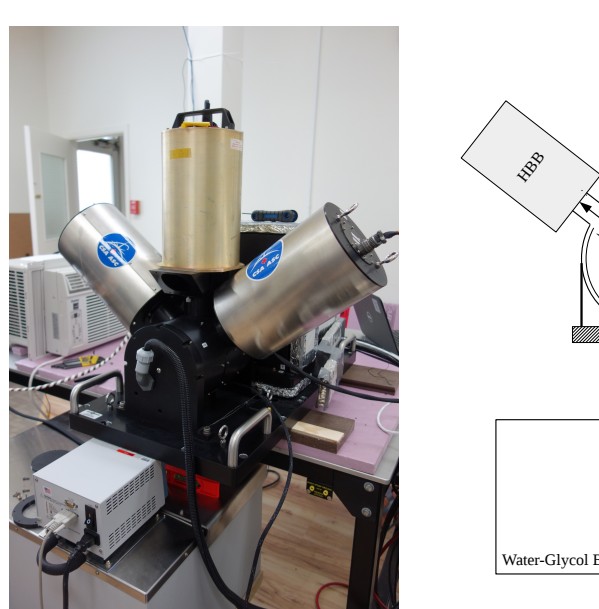
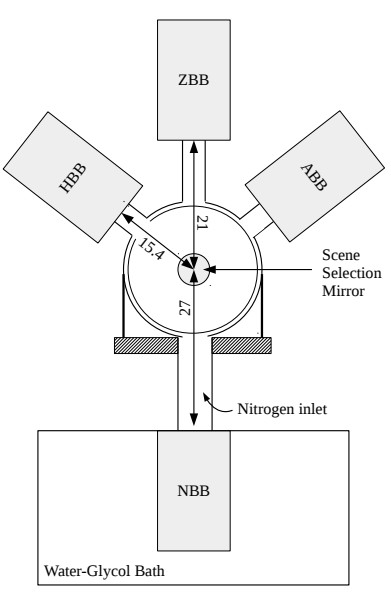

**Figure 4.** (left) Laboratory experimental setup for instrument characterization. The lateral blackbodies are those of the FIRR while that in zenith position was temporarily mounted. The white container at the bottom holds another blackbody immersed in a water-glycol bath regulated by a cryogenic system. (right) Schematic front view of the setup. The circle in the middle represents the scene selection mirror. Distances are in cm.




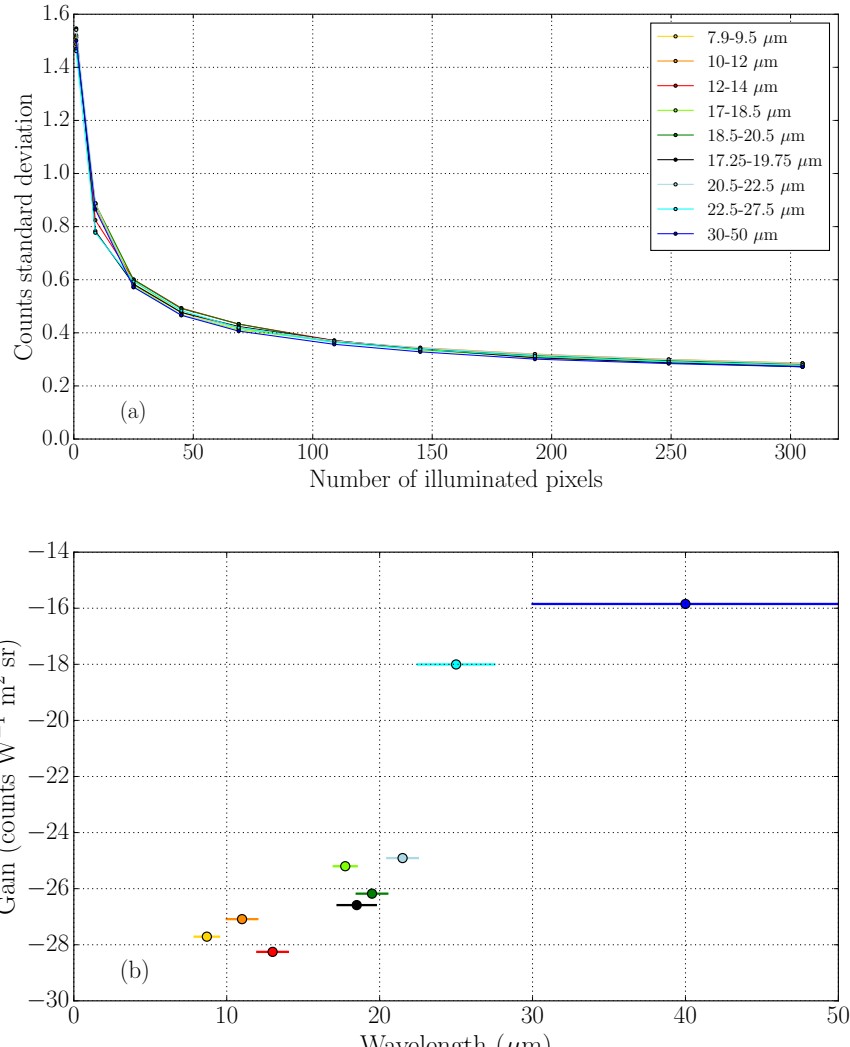

**Figure 5.** (a) Standard deviation of measurements performed on the ABB for different illuminated areas. The area are defined as disks around the central pixel. The average over 10 consecutive sequences is shown. (b) Average gain over the same 10 sequences for all bands. Horizontal bars highlight the spectral extent of the bands.




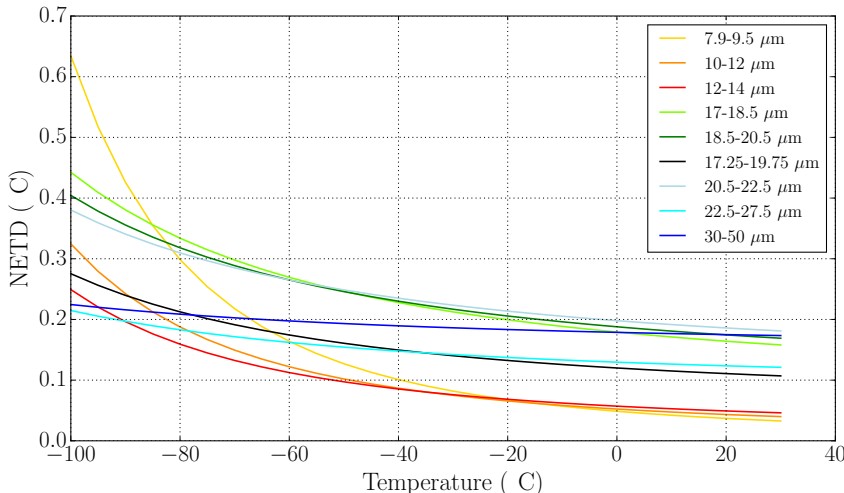

**Figure 6.** Noise equivalent temperature difference (NETD) corresponding to a noise equivalent radiance (NER) of 0.01 W m$^{-2}$ sr$^{-1}$, as a function of scene temperature for all spectral bands of the FIRR.

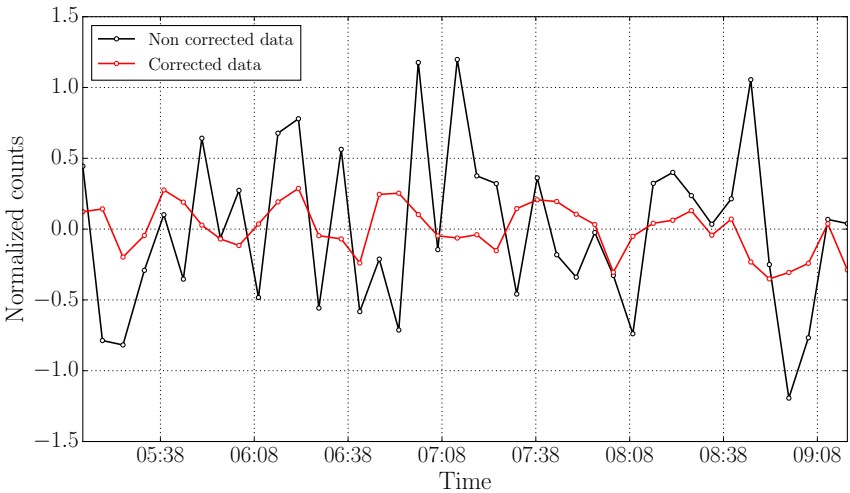

**Figure 7.** Average counts difference between measurements taken on ABB (T$_{ABB}$ $\simeq$ 25°C) and HBB (T$_{HBB}$ = 55°C) for 40 consecutive sequences. The black line was obtained without using the non-illuminated pixels while the red line corresponds to corrected data. All data were translated to null average for the sake of clarity.





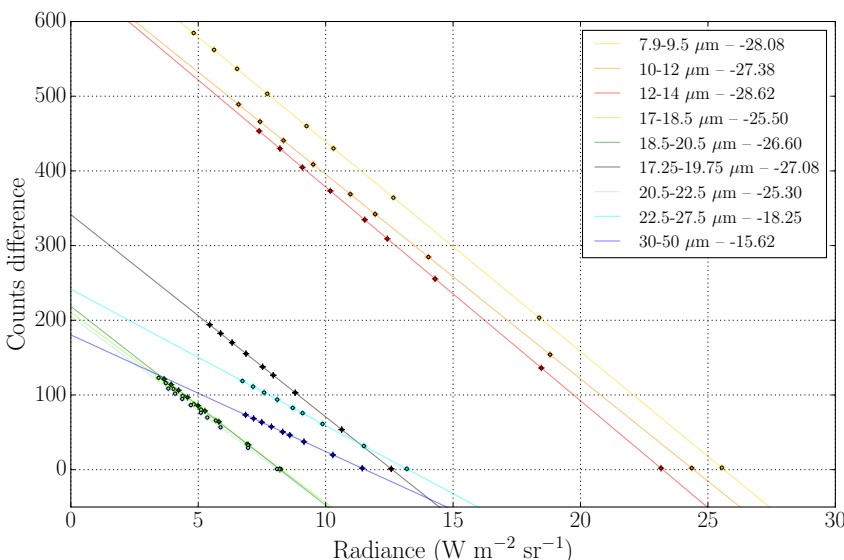

**Figure 8.** Counts difference between corrected signals on NBB and HBB in terms of NBB radiance in each spectral band of the FIRR. Dots indicate the average value for each temperature step. Individual measurements are depicted by the error bars. The lines are the linear regressions, whose slopes are indicated in the legend.





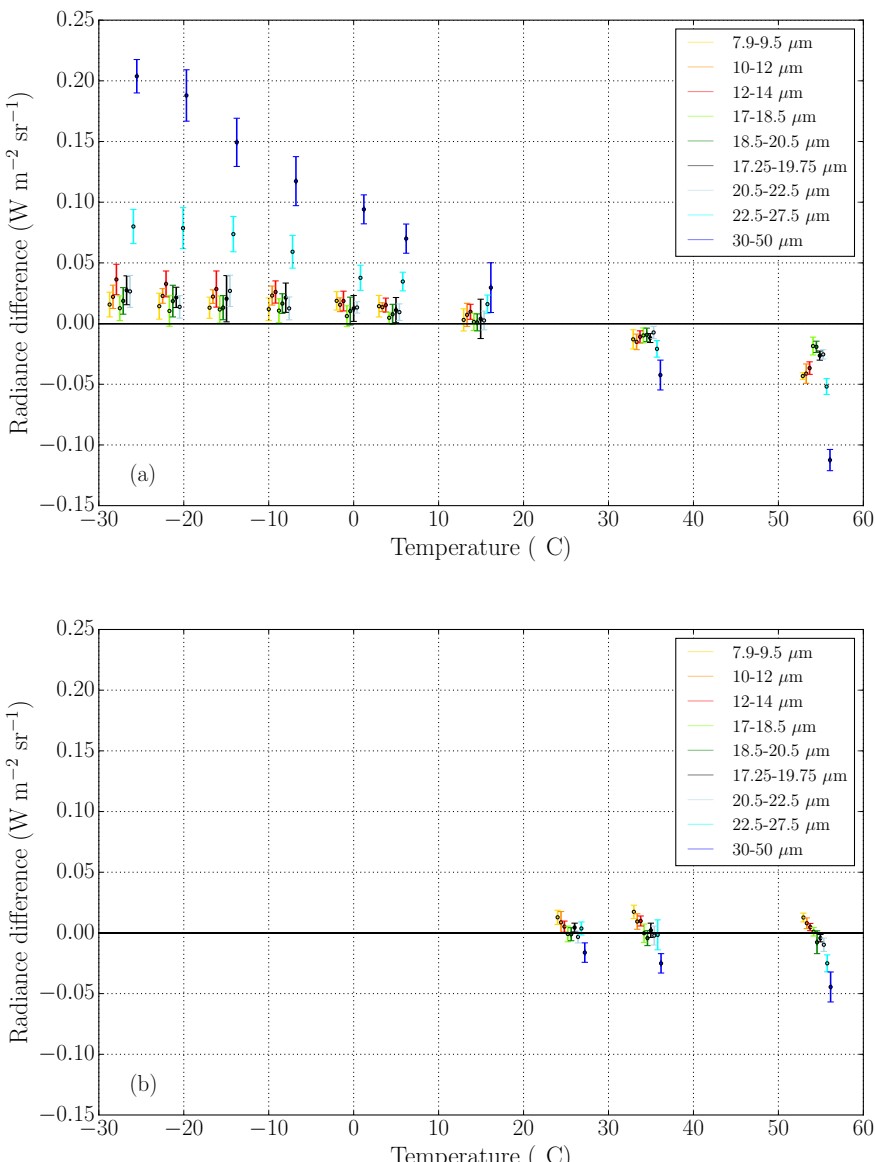

**Figure 9.** (a) Difference between NBB radiance retrieved with the FIRR algorithm and theoretical radiance for all NBB temperature steps. The error bars show all differences for a given step. (b) Same as (a) but for ZBB radiances.





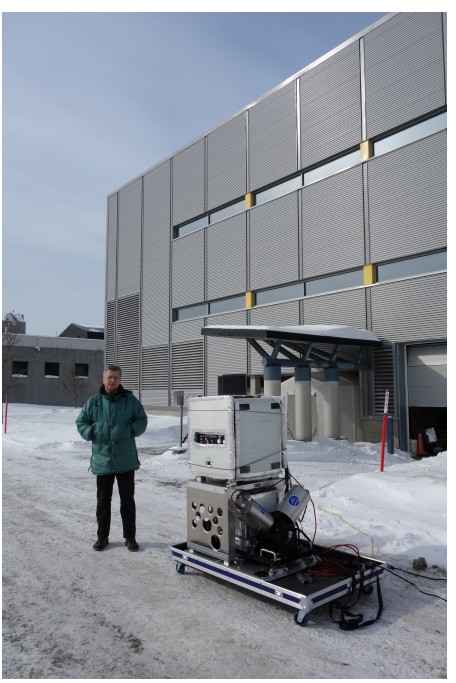

**Figure 10.** FIRR setup for ground operation at INO facilities on 21 February 2015.

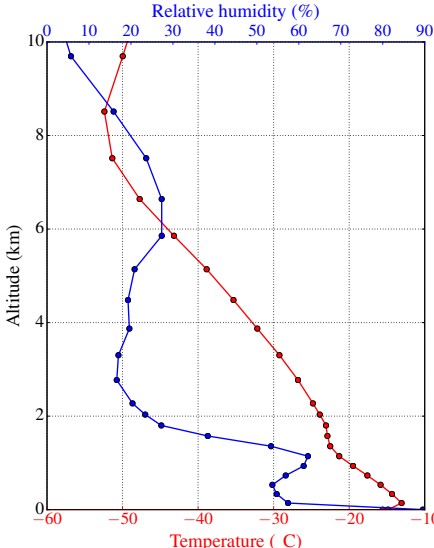

**Figure 11.** ERA-Interim profiles of temperature and humidity on 21 February 2015, 00:00 UTC. For the sake of clarity, the profile is shown up to 10 km only, but the reanalysis used for the radiative transfer calculations actually extends up to 46.2 km.





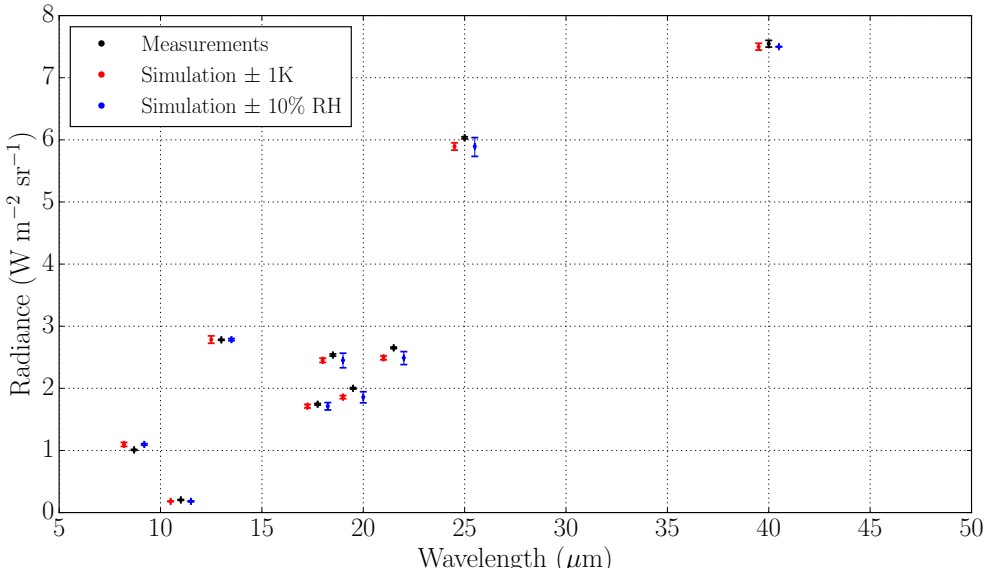

**Figure 12.** Radiances measured with the FIRR and simulated with MODTRAN for the 21 February 2015 night experiment. Blue error bars correspond to simulations with temperature profiles offset by ±1 K. Red error bars correspond to simulations with specific humidity profiles offset by ±10 %.

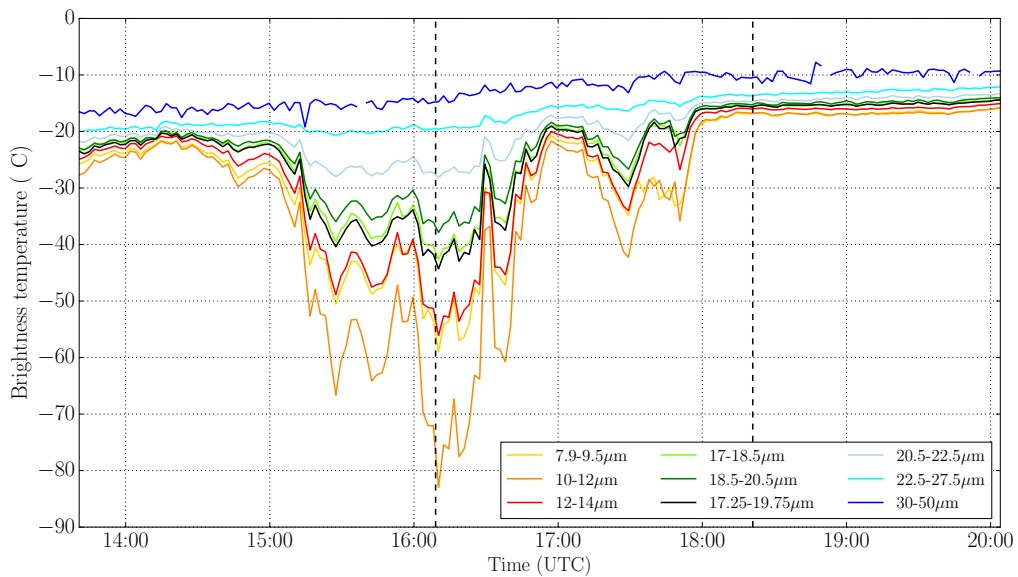

**Figure 13.** Time series of FIRR brightness temperatures for the 21 February 2015 experiment. Vertical dashed lines indicate when the 4 measurements shown in Fig.14 were taken.





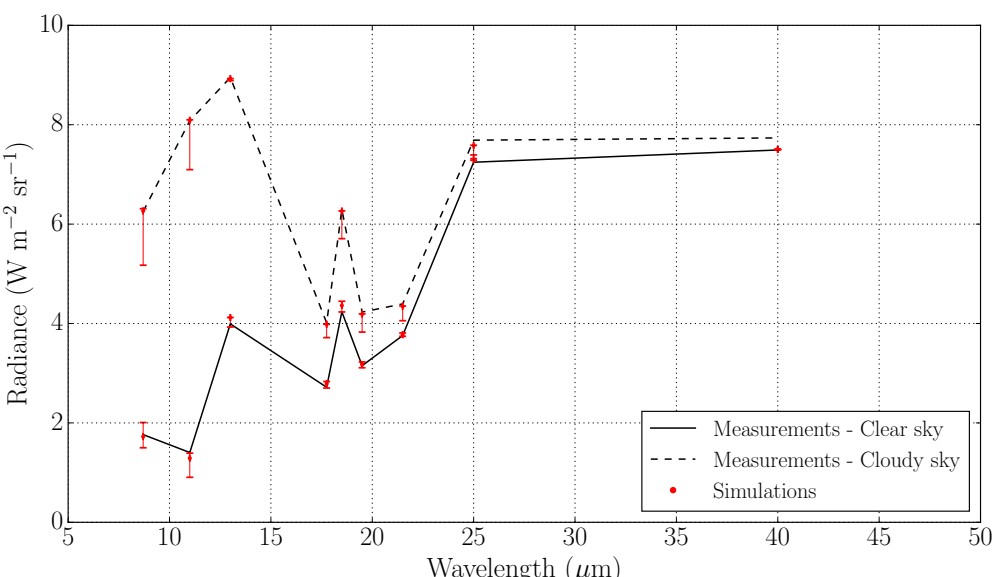

**Figure 14.** Radiances measured with the FIRR and simulated with MODTRAN for the 21 February 2015 experiment. Two cases are shown with different cloud covers. The clear sky simulation corresponds to a cloud between 2.3 and 3.8 km with $\tau = 0.3$ and $d_{\mathrm{eff}} = 20$ $\mu$m and the cloudy sky simulation corresponds to a cloud between 2.3 and 3.8 km with $\tau = 12.5$ and $d_{\mathrm{eff}} = 40$ $\mu$m. Error bars indicate the range of radiances obtained for $d_{\mathrm{eff}}$ varying from 3 to 250 $\mu$m.





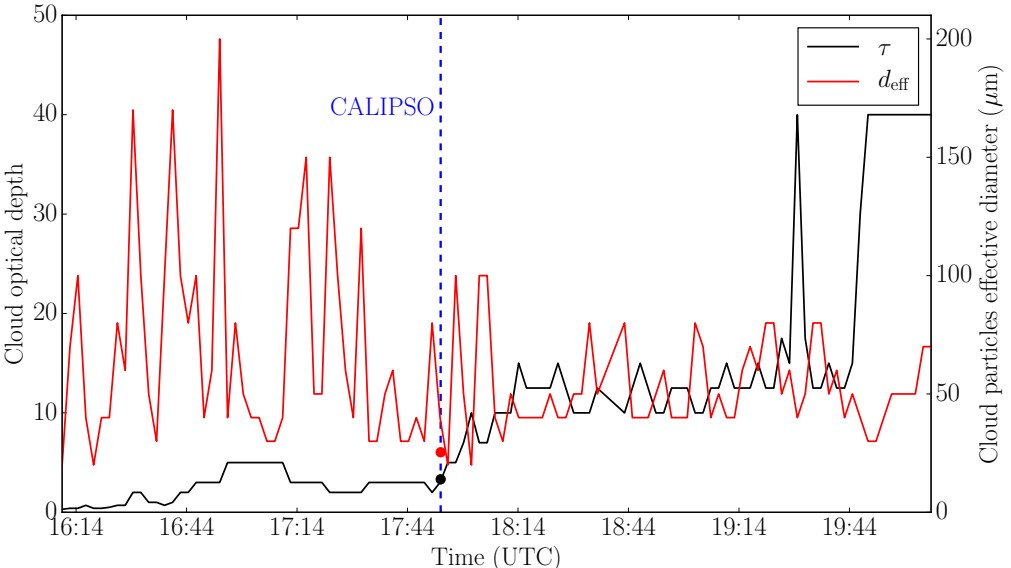

**Figure 15.** Time series of optical depth and effective diameter retrieved from FIRR measurements performed on 21 February 2015. The values retrieved from CALIPSO instruments 33 km East of the measurement site are indicated by large dots.

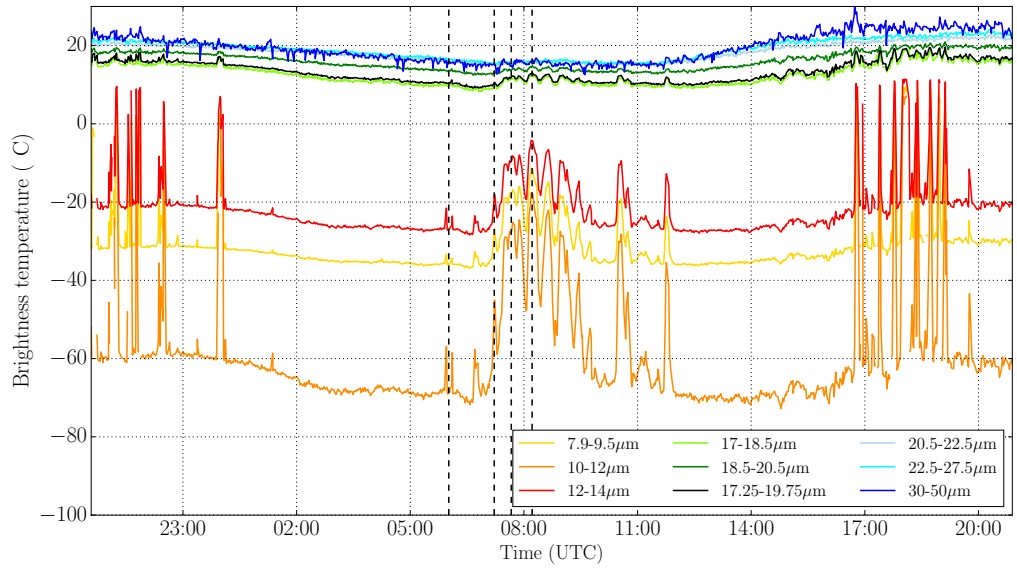

**Figure 16.** Time series of FIRR brightness temperatures for the 2 July 2015 experiment. Vertical dashed lines indicate when the 4 measurements shown in Fig.17 were taken.





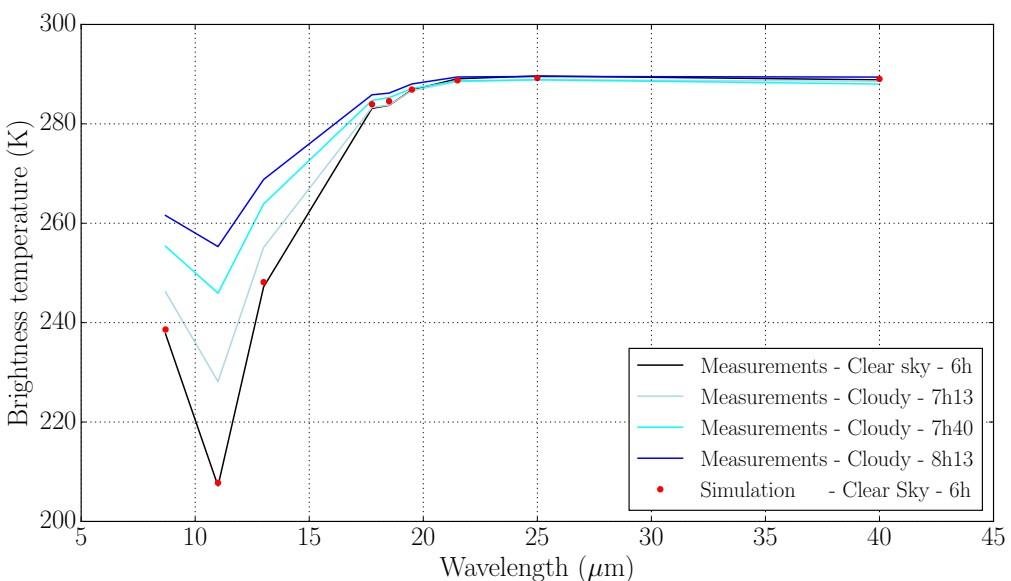

**Figure 17.** Brightness temperatures measured with the FIRR during the 2 July 2015 experiment. Four cases are shown with different cloud covers. They are also reported in Fig. 16. The MODTRAN simulation corresponding to the clear sky conditions is indicated by red dots.