# Peer review of "A microbolometer-based far infrared radiometer to study thin ice clouds in the Arctic"

_Atmospheric Measurement Techniques, 2015_

## Referee Comment (RC1) · B. J. Drouin (Referee) · 2 Feb 2016

Current prospects for FIR remote sensing from orbit are improving and a number of instrumental architectures may be useful for sensing in this spectral region. Unknowns in both clear and cloudy sky radiance/irradiance result in great difficulties for models that attempt to constrain the OLR which is generally highly variable. The TICFIRE mission will address uncertainties in FIR interaction with thin ice clouds through imaging measurements in 9 spectral bands from 8-50 microns. The development of TICFIRE instrumentation and its deployment for ground testing is detailed in this report. Although the mission is described as an imager, the demonstrated hardware does not meet this goal.

Although the instrument functioned in its capacity to measure FIR radiation, the perfor-

mance was limited due to the need for spatial averaging across the focal plane. Nevertheless, the results are still useful for statistical and modeling purposes and provide valuable insight into FIR cloud interactions.

It is not clear whether or not the TICFIRE mission will have to change its objectives to accommodate for the reduced sensitivity. The conclusion vaguely refers to the "ergodicity hypothesis", this should be discussed and quantified if the presented hardware is to be useful for orbital remote sensing.

Some technical aspects of the calibration procedure are noteworthy. (1) The filter wheel is said to have an 'opaque position' which is a curious thing considering the blocking object itself would have some blackbody temperature and emissivity that will produce a signal for FIR detectors; (2) similarly each spectral filter might be characterized for spectral emissivity as well as spectral transmission; (3) perhaps the blackbody calibration will account for some of the filters spectral emission, but there does not seem to be a wavelength dependent calibration associated with the filters themselves as radiative sources, and being situated some distance from the focal plane and filling the field of view, these photons may be expected to travel through the optics differently (compared to the transmitted scene) perhaps adversely affecting the image.
* * *

---

## Referee Comment (RC2) · Anonymous Referee #1 · 9 Feb 2016

This paper describes a new far-infrared radiometer that has been built and preliminarily deployed in support of the TICFIRE satellite project of the Canadian Space Agency.

This discussion paper is generally well-written and has a considerable amount of detail, including demonstration measurements that are very nice to have. It will therefore serve as a very useful resource for future development of far-infrared radiometry.

However, this discussion paper would benefit from a number of relatively minor, but still necessary, changes that should be incorporated prior to publication in AMT. These changes are listed below:

1. The spectral transmittance of the 9 filters needs to be discussed in considerably more detail. What materials were used? Was the choice of spectral response for each filter driven largely by the limitations in the materials, or were they chosen specifically
for science value? If it is the former, does more work need to be done to develop exotic (or mundane) materials that can be used for these filters? Can the spectral response for each filter be modified in future versions of this radiometer? Are the spectral responses stable, or will they degrade in unknown ways either on the ground, on an aircraft, or in a space environment? I am confused by the choice of the 10-12 $\mu$m band, as this appears to be partially contaminated by O3. Also, a figure is needed for atmospheric transmission to TOA vs height per band as a function of column water vapor.

2. It was not made clear to this reviewer if the plan is for the instrument, as described here, to be flown in TICFIRE, or if this is just a stepping-stone to the instrument that will be flown in TICFIRE. Is the idea to demonstrate this capability on a breadboard and then build something identical that is space-qualified? Are the components of the radiometer described here space-qualified, or is additional technology demonstration required?

3. Can the change in emissivity of the blackbodies over time be estimated? Can this change be estimated on orbit? Is it important, or are there a large number of internal reflections so it doesn't really matter?

4. What is meant by scene temperature for Figure 6? For an observation where each filter is giving a different brightness temperature reading, should the reader expect to use Figure 6 as an estimate of each filter's NETD based on that filter's brightness temperature?

5. The authors should comment on whether TICFIRE will be able to see the effects of surface processes on dehydrated conditions? Recent publications (e.g., Chen et al, GRL, 2014 and Feldman et al, PNAS, 2014) have highlighted the large differences in far-IR surface emissivity between frozen and unfrozen surfaces, with large scientific implications for polar feedbacks. Would any of the filters be able to reliably detect a signal arising from a difference in far-IR surface emissivity of 0.1? of 0.2?

---

## Author Comment (AC1) · 30 Mar 2016

**Response to B. J. Drouin (reviewer 1)**

The reviewer's comments are in black and our answers are in red.
Modifications of the manuscript are reported in bold and italic.
The pages and lines reported here correspond to the original pdf.

Current prospects for FIR remote sensing from orbit are improving and a number of instrumental architectures may be useful for sensing in this spectral region. Unknowns in both clear and cloudy sky radiance/irradiance result in great difficulties for models that attempt to constrain the OLR which is generally highly variable. The TICFIRE mission will address uncertainties in FIR interaction with thin ice clouds through imaging measurements in 9 spectral bands from 8-50 microns. The development of TICFIRE instrumentation and its deployment for ground testing is detailed in this report.

We thank B. J. Drouin for reviewing this work and provide below a point-by-point reply to his comments.

Although the mission is described as an imager, the demonstrated hardware does not meet this goal. Although the instrument functioned in its capacity to measure FIR radiation, the performance was limited due to the need for spatial averaging across the focal plane. Nevertheless, the results are still useful for statistical and modeling purposes and provide valuable insight into FIR cloud interactions.

The FIRR was not meant to be an imager, it is rather a preliminary version of TICFIRE aimed at testing the detector, the filters and the impact of atmospheric properties on its FIR signature. The TICFIRE instrument design will be different in many ways, including the optics (larger fov), microbolometers (faster response for higher acquisition rate) and potentially the filters (higher transmittance). We have now clearly stated along the manuscript that the FIRR and TICFIRE instruments are different:

p.1, l.1:
"A far infrared radiometer (FIRR) dedicated to measure radiation emitted by clear and cloudy atmospheres was developed  *in the framework of* the Thin Ice Clouds in Far InfraRed Experiment (TICFIRE) *technology demonstration* satellite project."

p3. l.69:
"Here we present the far infrared radiometer (FIRR) prototype designed to measure radiation in 9 spectral bands ranging from 8 to 50 μm. *The FIRR is aimed at demonstrating the capability of a microbolometer-based radiometer to accurately measure F-IR radiation*. The design and data acquisition procedure [...]"

p.8, l.237:
"*Contrary to TICFIRE that will have a much larger field of view and significantly different optics*, the FIRR is not intended to be used as an imager. *Hence* the calibrated radiances are averaged [...]"

p.15, l.482:
"As the TICFIRE is meant to be an imager, spatial averaging will be limited. *In addition, since the optics and detectors of TICFIRE will be different than those of the FIRR, it is hazardous to apply FIRR radiometric characteristics to TICFIRE. However, it has been shown that spatial averaging over a reduced number of pixels, as well as temporal averaging, could improve TICFIRE radiometric performances.* All in all, a *trade* will have to be made [...]"

Although FIRR and TICFIRE are different, the results obtained with the FIRR should be used as much as possible to provide information about the TICFIRE capabilities. For this reason, we extented the analysis shown in Fig. 7 to compute the standard deviation of the corrected signal for different spatial averages (= different numbers of illuminated pixels used for the average). These results are presented in Fig. 1. This figure was not added to the manuscipt, but the latter was modified as follows:

p.10, l.306:
"*To quantify the impact of spatial averaging on FIRR resolution the same analysis was performed for reduced numbers of illuminated pixels. When a single pixel is used, the standard deviation of the corrected signal is 0.75 counts, but it drops to 0.45 when 2 pixels are used, and 0.4 when 4 pixels are used. In view of the TICFIRE imaging application, this proves that spatial averaging over a limited number of pixels could significantly increase the radiometric resolution, even though this would be at the expense of spatial resolution*."

[Figure]

*Figure 1: Standard deviation of the count difference between corrected measurements taken on ABB and HBB (filter 7.9-9.5 μm) over 40 consecutive measurements, as a function of the number of illuminated pixels used for the spatial average.*

It is not clear whether or not the TICFIRE mission will have to change its objectives to accommodate for the reduced sensitivity. The conclusion vaguely refers to the "ergodicity hypothesis", this should be discussed and quantified if the presented hardware is to be useful for orbital remote sensing.

The evasive reference to the ergodicity hypothesis was removed (see above) to stress that FIRR performances cannot be straighforward extrapolated to TICFIRE performances, especially because the optics and detectors of TICFIRE will be to some extent different than those of the FIRR. The design of TICFIRE is still under discussion at the Canadian Space Agency with the industrial partners and the results presented here are intended to help with the decisions relative to TICFIRE design:

p.16, l.515:
"These preliminary results nevertheless represent a substantial step toward the TICFIRE mission, *whose design is currently discussed with industrial partners at the Canadian Space Agency.*"

Some technical aspects of the calibration procedure are noteworthy.

(1) The filter wheel is said to have an 'opaque position' which is a curious thing considering the blocking object itself would have some blackbody temperature and emissivity that will produce a signal for FIR detectors;

Even though the utility of this opaque position is not detailed in the manuscript it mainly serves two objectives. 1) It insulates the optomechanical device from the outside air to prevent humidity or dust to get in. 2) It is used to distinguish the contributions of the blackbodies enclosure (including scene selection mirror) from that of the filters to the total background signal. This latter point is critical for the future improvement of the calibration algorithm, especially for measurements taken in harsh environmental conditions.

p.4, l.100:
"[...]as well as an opaque position and an open position, *the last two positions being essentially used to investigate the thermal behavior of the filters and calibration enclosure*."

(2) similarly each spectral filter might be characterized for spectral emissivity as well as spectral transmission;

So far, the filters emissivity was not characterized, only their transmittance was. At first order this is nevertheless not critical because the contribution of the filters to the background signal can be removed using the blackbodies calibration. This holds at least if their temperature remains constant throughout a measurement sequence, which was essentially the case during the experiments presented in the paper and performed in well-controlled environments or calm environmental conditions.

Knowledge of the emissivity becomes more important when filter emission can vary at time scales shorter than the calibration procedure, in which case the filters contribution to the background is not entirely removed with the blackbodies calibration. However, in such case the radiative temperature of the filter should be known accurately as well. This is practically very difficult because this very temperature is not measured and seems (ongoing detailed analysis of in situ measurements) to be significantly different than the temperatures recorded at various locations within the optomechanical device.

p.6, l.168:
"This contribution *can not be accurately estimated because the radiative temperature of the filter is unknown, and it* can not be removed by the calibration procedure, making it particularly critical."

(3) perhaps the blackbody calibration will account for some of the filters spectral emission, but there does not seem to be a wavelength dependent calibration associated with the filters themselves as radiative sources, and being situated some distance from the focal plane and filling the field of view, these photons may be expected to travel through the optics differently (compared to the transmitted scene) perhaps adversely affecting the image.

We do not think that the filters require a wavelength-dependent calibration for their radiative emission.

In fact emission from the filters themselves is not different from emission from any other component of the enclosure that is in the field of view of the detector. As detailed above and in the manuscript (p.5, l.162), for the calibration procedure to remove the emission of the filters along with emission from the other components, the filters temperature should always be similar to that of the other components. If this is not the case, then the calibration is imperfect, as mentioned in the manuscript (p. 6, l. 168).

---

## Author Comment (AC2) · 30 Mar 2016

**Response to reviewer 2**

The reviewer's comments are in black and our answers are in red. Modifications of the manuscript are reported in bold and italic. The pages and lines reported here correspond to the original pdf. New references can be found at the end of the document.

This paper describes a new far-infrared radiometer that has been built and preliminarily eployed in support of the TICFIRE satellite project of the Canadian Space Agency. This discussion paper is generally well-written and has a considerable amount of detail, including demonstration measurements that are very nice to have. It will therefore serve as a very useful resource for future development of far-infrared radiometry. However, this discussion paper would benefit from a number of relatively minor, but still necessary, changes that should be incorporated prior to publication in AMT. These changes are listed below:

**We are grateful to this reviewer for the encouraging comments, and tried to add the pieces of information needed to clarify the points raised. These modifications are detailed below.**

1. The spectral transmittance of the 9 filters needs to be discussed in considerably more detail. What materials were used? Was the choice of spectral response for each filter driven largely by the limitations in the materials, or were they chosen specifically for science value? If it is the former, does more work need to be done to develop exotic (or mundane) materials that can be used for these filters? Can the spectral response for each filter be modified in future versions of this radiometer? Are the spectral responses stable, or will they degrade in unknown ways either on the ground, on an aircraft, or in a space environment? I am confused by the choice of the 10-12  $\mu$ m band, as this appears to be partially contaminated by O3. Also, a figure is needed for atmospheric transmission to TOA vs height per band as a function of column water vapor.

According to the reviewer, additional technical details about the filters are given below. Many details are provided here but all those details are not added to the manuscript.

Initially the instrument was supposed to host 6 filters (1, 2, 3, 6, 8, 9 of Table 1). The 6 spectral bands were chosen for their scientific value, with the intent to cover as much as possible the whole spectrum from 8 to 50  $\mu$ m (excluding the CO2 band). The width of the bands was chosen so that a sufficient and relatively constant amount of radiation lies in each band. Later on the budget of the project was increased, allowing us to buy 3 new filters. It was decided to split the band 6 in 3 different bands (4, 5 7) because it corresponds to the spectral region where the atmospheric cooling rates greatly vary with altitude (see Plate 6 of Clough et al., 1992). Increasing the spectral resolution of the FIRR in this region should provide a better vertical resolution of the retrievals.

Based on these scientific requirements the 9 filters were ordered to 4 different commercial companies (Reynard Corporation, University of Reading, Infrared Filters Solutions, QMC Instruments) used to providing filters for spatial applications (note that only few companies are able to design bandpass filters in the far-infrared, which limited our choice). The final spectral response of each filter is the result of a trade between the science value (requirement) and technological limitation. For instance, the leak at 31  $\mu$ m for the band 17 – 18.5  $\mu$ m (which was erroneously missing on the dicussion paper) is a limitation of the material used for the coating (Fig. 1). Getting rid of this leak would have implied a reduction of the overall transmittance of the filter, which would have been even more critical.

Figure 1: Spectral transmittances of FIRR filters updated with the leak at 31  $\mu$ m.

The substrate for filters 1 and 2 is Zinc Sulfide. The substrate for filters 3 is Zinc Selenide. The substrate for filters 4, 5, 6, 7, 8 is Cadmium Telluride. We do not know the material of the mesh filter 9, neither the coatings used for the interference filters.

**The details regarding the filters suppliers and materials have been added to Table 1.**

So far the Canadian Space Agency does not intend to invest much into improving the filters and prefers using existing technology. Similar filters were already used for the Mars Climate Sounder on the Mars Reconnaissance Orbiter (McCleese et al., 2007). The effort is rather put on the microbolometers sensors developed at INO, which constitute the originality of the FIRR instrument. Nevertheless, it remains possible to change the filters for the future versions of the radiometer (including that on the TICFIRE satellite) according to the results of the ongoing experiments.

The filters were delivered around December 2014 and their transmittances were measured at this time. Since then they are in the instrument and have not been characterized again. However, the measurements taken in the last year do not show any indication of filters degradation and the companies mentioned above have worked in the past on satellite projects so that we are confident about the robustness of the filters. We intend to measure the filters transmittance in the course of the summer 2016, which will provide information about their degradation.

Regarding the overlap of the FIRR bands with the 9.6  $\mu$ m absorption band of ozone, we first point out that it is rather the band 7.9 – 9.5  $\mu$ m that is impacted as shown in Figure 2. This overlap was initially not considered as a problem and when we actually realized that it could be one, the filter was already delivered and it was too late to order a new one. This filter will be modified for the future versions of the radiometer to avoid this overlap.

*Figure 2: Transmittances of the FIRR filters 1 and 2, and atmospheric transmittance, highlighting the ozone absorption band.*

As suggested, a figure showing the transmittance of the atmosphere in the FIRR bands as a function of column water vapor was added (Fig. 3).